# A New Ten-Step Surgical Approach to Mayer–Rokitansky–Küster–Hauser Syndrome—A Preliminary Report of Three Cases

**DOI:** 10.3390/jcm14041136

**Published:** 2025-02-10

**Authors:** Valentin Nicolae Varlas, Yassin Rhazi, Roxana Georgiana Varlas, Hamza Ouzaher, Benyounes Rhazi

**Affiliations:** 1Department of Obstetrics and Gynecology, Filantropia Clinical Hospital, 011132 Bucharest, Romania; valentin.varlas@umfcd.ro; 2Department of Obstetrics and Gynecology, “Carol Davila” University of Medicine and Pharmacy, 37 Dionisie Lupu St., 020021 Bucharest, Romania; 3Clinic Achark, Rue Saad Zaghloul, Oujda 60000, Morocco; ouzaherhamza@gmail.com (H.O.); rhazi-benyounes@hotmail.fr (B.R.)

**Keywords:** Mayer–Rokitansky–Küster–Hauser syndrome, vaginal agenesis, neovagina, sacropexy of neovagina, modified Davydov procedure, surgical steps video

## Abstract

**Background:** Vaginal reconstruction procedures for patients with Mayer–Rokitansky–Küster–Hauser syndrome (MRKH) have the main purpose of restoring the anatomy to increase the quality of life of these patients. To describe the surgical treatment of patients with type I Mayer–Rokitansky–Küster–Hauser (MRKH) syndrome with complete vaginal agenesis in 10 steps, using a sacropexy technique by a double approach (laparoscopic and perineal), which could help make this procedure more accessible and safer. **Methods:** The surgical technique was used in a group of three patients diagnosed with MRKH syndrome with vaginal agenesis, in which a neovagina with peritoneal flaps was created, and the reconstruction of the vaginal apex and its sacropexy created the conditions for a favorable and lasting result. **Results:** Annual reevaluations for up to 5 years revealed a functional neovagina with an average length of approximately 11.33 cm, without stenotic aspects, and no granulation tissue formation. All three cases in which this technique was performed reported sexual activity as expected, excellent quality of life, and good psycho-emotional reintegration. It should be noted that two of the three patients also resolved their marital situation. **Conclusions:** Although the number of patients in this preliminary report is limited, the surgical technique presented is an effective, safe approach with very good anatomical and functional results at the 5-year follow-up. The favorable surgical outcome of these cases also determined the social integration of the patients, solving some ethnic and religious problems.

## 1. Introduction

Mayer–Rokitansky–Küster–Hauser syndrome (MRKH) is a congenital disorder characterized by hypoplasia or aplasia of the uterus and vagina, with ovaries with preserved function and the presence of normal secondary sex characteristics. These patients’ karyotype (46, XX) and phenotype are normal [1,2]. The etiology of MRKH syndrome is still difficult to identify; it can be polygenic with autosomal dominant transmission, epigenetic, the penetrance of a gene mutation, multifactorial, or due to environmental disruptors (phthalates) [2]. The possible genes involved in the etiology of MRKH are WNT4, HOXA, RBM8A, TBX6, LHX1, and HNF1B [3]. The prevalence of MRKH syndrome is approximately 1 in 5000 live female newborns [4].

Müllerian agenesis is classified into type I MRKH syndrome, characterized by uterine and upper two-thirds of the vagina aplasia, and type II MRKH syndrome, which is associated with extra-Müllerian anomalies with evidence of multiple morphological anomalies, best described as the MURCS complex—Müllerian duct agenesis, renal agenesis, and cervicothoracic somite anomalies, with a rate of 43.5–56.5% [5]. The incidence of renal anomalies (unilateral renal agenesis, ectopic kidney, and horseshoe kidney) varies between 13.1 and 34.2%, skeletal anomalies (Klippel–Feil anomaly and scoliosis) between 10 and 39.2%, auditory anomalies (hearing loss/deafness) between 5 and 11%, cardiac anomalies <5%, neurological anomalies, and ovary anomalies [2,4,6].

The diagnosis and treatment of infertility and obstetric management require a classification that is as accurate as possible in terms of the assessment of individual anatomical compartments. Thus, Kiblboeck et al., in the multicenter prospective study EVA (ESHRE/ESGE—VCUAM—AFS), showed better results regarding the organ-based ESHRE/ESGE and VCUAM classifications compared to the AFS classification [7].

The treatment of this condition is a touchstone for gynecological surgery, with psychosocial, cultural, and religious implications, which can affect the quality of life of these patients [8,9]. Over time, several surgical and non-surgical techniques have been presented, without defining the procedure with the best anatomic–functional result. The best technical procedure for a vaginoplasty is chosen after careful patient counseling, and depends on the patient’s condition and the surgeon’s experience. The success of creating a neovagina with appropriate dimensions consists of low complication rates, good cosmetic results, good quality of life, and satisfactory sexual function. Vaginal reconstruction requires the restoration of the anatomy by creating a neovagina that is permeable to two fingers and has a length of at least 6 cm at 6–12 months postoperatively, while functional resolution is assessed by the patient’s satisfaction with sexual activity [10].

A comparative evaluation of non-surgical procedures (vaginal dilation) with surgical ones revealed a reduction in the length of the postoperative neovagina in the first group, without any differences in sexual function or quality of life [11]. Among the surgical interventions, the most common is the McIndoe procedure, but the highest anatomical and functional success rates have been recorded with the Davydov and Vecchietti procedures. The Davydov procedure is less painful and is associated with the achievement of a longer vagina, while the Vecchietti technique is minimally invasive and has long-lasting effects [12].

The most used laparoscopic approach is the Davydov procedure, which creates a neovagina by expressing the peritoneum and anchoring it to the vaginal introitus through a series of sutures. This procedure has several advantages and disadvantages. Among the advantages, we have a fast, safe technique, with good scarring and good functional status. However, among the disadvantages, we have an inadequate length of the neovagina, a risk of stenosis in the upper third, and prolapse [13].

Pennesi et al., in a cross-sectional study using an online survey of 614 patients with Mayer–Rokitansky–Küster–Hauser syndrome from 40 countries, found that 54% had vaginal lengthening as a therapeutic indication; 39% of these patients benefited from a surgical correction. Of these patients, 70% reported urinary symptoms (urinary incontinence, frequent urination, urinary urgency, and recurrent infections) and symptoms of prolapse (lower abdominal pressure [41%], pelvic heaviness [29%], and vaginal swelling [11%]) [14].

Our surgical procedure, unlike other surgical techniques, addresses the prophylaxis of urinary and pelvic floor symptoms (neovaginal apex prolapse). The presented laparoscopic procedure derives from the Davydov technique and consists of creating a neovagina by mobilizing and pulling down the pelvic peritoneum, anchoring the peritoneal flaps to the newly created (in vaginal agenesis) or pre-existing (in vaginal hypoplasia) introit, followed by the sacropexy of the neovaginal apex with a synthetic mesh.

MRKH syndrome is often diagnosed in adolescence with primary amenorrhea. Counseling for these patients is very important because there is a significant psychological impact regarding one’s own identity and the recognition of sexuality, as well as problems related to the desire to conceive [15]. The key to solving this situation is the creation of a functional neovagina that allows the patient to have proper intercourse and eventually attain marital status through social integration [2].

These patients’ quality of life and social integration are affected secondary to the lack of a uterus and vagina, making it impossible to perform normal vaginal intercourse. Regarding the ability to conceive, uterine transplant surgical procedures have been successfully performed [1,2].

Below, we present a series of three cases with MRKH syndrome subjected to the new technique of perineal vaginoplasty with laparoscopic sacropexy of the neovaginal apex with synthetic mesh, in which the immediate follow-up and that 5 years later showed that the anatomic–functional and psychosexual results were favorable.

## 2. Patients and Methods

### 2.1. Patients

Over 2 years (between January 2016 and September 2017), 3 patients with type I MRKH syndrome underwent perineal vaginoplasty technique and laparoscopic neovaginal sacropexy in the Achark Clinic of Obstetrics and Gynecology of Oujda, Morocco. Primary amenorrhea was the main clinical symptom of these patients. MRKH syndrome was diagnosed based on an ultrasound examination and the chromosomal karyotype. Patients with MRKH have no imbalances in sex steroid hormone levels or chromosomal abnormalities. One patient presented with vaginal hypoplasia, and two patients with vaginal aplasia. All counseled patients gave their informed consent before the surgical procedure.

### 2.2. Surgical Technique

The surgical procedure had a dual approach (perineal/vaginal and laparoscopic), the reconstruction of the neovagina in patients with MRKH syndrome with vaginal aplasia or hypoplasia, being performed in several operative steps (10-step algorithm) (Table 1). A neovagina with peritoneal flaps was created. The reconstruction of the vaginal arch and its suspension from the sacred band created the premise for a favorable and lasting result, and decreased risk of possible vaginal prolapse.

The perineal approach consisted of either making a transversal suburethral incision between the small labia of approximately 3 cm on the vaginal mucosa at the level of the pre-existing introit or an incision at the level of the future vaginal introit (Figure 1a and Figure 2a). To facilitate the dissection of the rectovesical space, a saline solution with adrenaline (dilution 1:200,000) was injected with a long needle. Subsequently, dissection was performed in the rectovesical space through a sharp incision, which was then extended in depth. A compress using forceps was then introduced into the newly created space between the bladder and rectum (Figure 1b and Figure 2b). In the upper portion of the newly created cavity, the dissection was performed with laparoscopic assistance to prevent the risk of injuring the adjacent organs (bladder, bowel, and rectum).

Then, a horizontal incision of the apex of the peritoneal mucosa was made. The peritoneum of the pouch of Douglas was pulled down through this artificially created channel, the apex of the peritoneal mucosa was horizontally incised, and then the lateral peritoneal flaps were sutured with separate 2-0 Vicryl threads to the mucosa of the neovaginal introit (Figure 1c,d and Figure 2c). The apex of the neovagina was then anchored to the periosteum of the sacrum to avoid prolapse of the vault of the neovagina (Figure 1k and Figure 2d). At the end of the intervention, synthetic dilators were inserted into the neovagina. Operation time was about one hour, with minimally intraoperative blood loss. The patients were hospitalized for three days, followed by ambulatory care for another 10 days.

At the end of the operation, a dilator covered with a condom was inserted into the neovagina, covered with an oily solution to make it less painful. Povidone–iodine cream or tetracycline ointment was applied to the dilator for infection prophylaxis.

## 3. Results

The data regarding the demographic characteristics of the patients, the surgical procedure, and the follow-up, respectively, are summarized in Table 2. An inspection of the peritoneal cavity revealed the absence of the uterus and the presence of two rudimentary horns; the ovaries and fallopian tubes were macroscopically normal. None of the patients presented with associated congenital anomalies of the kidneys, skeleton, or recto-anal region.

No intraoperative complications were recorded. The average operating time was 56 min, and the average intraoperative blood loss was <50 mL. Mean neovaginal length and width at discharge were 14.66 and 2.9 cm, respectively. The average duration of stay in the hospital was 3 days.

Postoperatively, the dilator was kept in for 24 h, and later, the patients used the dilator daily for approximately 24 h over 14 days. Later, the patients had sexual intercourse, and the initiation of the epithelialization process of the neovagina was also observed (Appendix A).

The operated patients had a favorable evolution, and were advised on how to use the dilators. The use of dilators is very important to keep the neovagina functional by preserving an appropriate length and avoiding the occurrence of vaginal stenosis. The patients used dilators daily for 30 min to facilitate intercourse after the neovaginal epithelization.

The assessment of the postoperative results was carried out by a visual inspection of the neovaginal mucosa and by measuring its diameter and length. In patients with MRKH syndrome, to evaluate a method’s success rate (sexual satisfaction), the FSFI (Female Sexual Function Index) questionnaire score is recorded. In these three patients, the scores obtained for the FSFI questionnaire highlighted a good and very good quality of sexual life one year after the reconstructive intervention (Table 3).

Some studies have also described normal scores but with pain in the lower third of the vagina and changes in the degree of lubrication [2,5]. The prediction of sexual dysfunction is indicated for a total FSFI score < 26 [16]. Women with MRKH syndrome have shown varying degrees of predictability regarding sexual dysfunction by reporting a low pain scale score, a negative genital self-image, and low sexual self-esteem. It is worth noting, however, that the total FSFI score for women with MRKH did not differ significantly from that for the control group [17].

The follow-up for these patients after surgery was every 3 months in the first year and then every 6 months for the next 5 years, with therapeutic success evidenced by an anatomical evaluation of the neovagina (the mean length being approximately 11.3 cm, with complete epithelialization). All three patients had a sexual partner, and two of them were married.

## 4. Discussion

Since the ideal surgery to reconstruct a neovagina has not yet been created, patient counseling continues to be very important for patients with MRKH syndrome. The success rate of the operation takes into account the surgeon’s experience; their willingness to try new surgical techniques; the patient’s confidence in the success of the operation; their geographical, ethnic, and religious characteristics; and the subsequent therapeutic management of these patients.

The diagnosis of MRKH is initially made by a transabdominal ultrasound and subsequently by an MRI evaluation to identify other possible associated malformations (renal, cardiac, and vertebral) that may accompany this condition. The karyotype is normal (46, XX), as are all hormonal levels, consistent with our case series [2].

The operative moment of performing a vaginoplasty depends on several social and economic factors, psycho-emotional maturation, and the desire to get married and to have a normal sexual life. The choice of the moment must be carefully identified because a very young age, the lack of a stable relationship, inadequate postoperative monitoring, minimal support from family, and incomplete treatment can lead to a major therapeutic failure.

Women diagnosed with varying degrees of MRKH syndrome have, as their primary goal, the desire to improve the quality of their sex life. The lack of clear guidelines regarding the benefits and possible complications related to the different treatment methods makes recommending a treatment regimen extremely difficult, requiring careful counseling based on the particularities of each case.

In patients with MRKH, congenital uterovaginal aplasia requires either the use of non-surgical self-dilation procedures, surgical dilation procedures, or skin or intestinal transplantation to create a neovagina [11,18]. Non-surgical vaginoplasty procedures are non-invasive, safe, do not require hospitalization, have low complications, low patient compliance related to the need for daily use of dilators, and have a success rate of 74–95% [19,20]. The non-surgical technique manages to achieve the best relationship between the anatomic–functional result and sexual satisfaction over time.

Patients strongly motivated by the desire to have sexual intercourse and who present with a vaginal pouch can use the conservative procedure described by Frank in 1938, which consists of self-dilation by using molds with a progressively increasing diameter that push the mucosa inward, and which has a success rate of 64% [21]. This technique can be used alone or before a surgical procedure.

In both conservative (self-dilation) and surgical dilation techniques, as well as postoperatively after a neovaginoplasty, specially designed molds are used, with progressive dimensions and different consistencies. In the first case, rigid molds were indicated; soft or semi-rigid molds were indicated in the other two cases. Their role is compressive, preventing postoperative hematomas, postoperative drainage, and preventing stenosis [20].

If initial, conservative methods fail, various surgical methods are indicated to achieve a functional and sensitive neovagina. According to the ACOG, vaginal dilation techniques are the first line in cases of vaginal hypoplasia, followed by the Vecchietti procedure as a second-line option [8]. Unlike the non-surgical ones, the surgical procedures currently used are invasive, relatively fast, associated with a higher rate of complications, have better results from a functional point of view, and have a lower rate of neovaginal tip prolapse.

The most commonly indicated surgical techniques for neovaginoplasty either use progressive dilation after performing rectovesical dissection (Vecchietti procedure [22]) or differ depending on the type of material used (skin graft—McIndoe procedure; peritoneum/amnion/intestines—modified McIndoe procedure; autologous peritoneal flap—Davydov procedure [12]; bowel—Baldwin technique [23]; and sigmoid colon). The latter two techniques require a transabdominal approach, which may increase patient morbidity.

The low incidence of MRKH syndrome has not allowed for more meta-analyses or randomized clinical trials to more accurately evaluate the therapeutic options. In a prospective randomized study by Cao et al. of a group of 40 patients with MRKH, the superiority of a laparoscopic peritoneal vaginoplasty (Davydov procedure) compared to a laparoscopic sigmoid colovaginoplasty (Baldwin procedure) was observed regarding the anatomical and functional results and satisfaction during sexual intercourse, respectively, without any difference in the average length of the neovagina or the rate of sexual satisfaction [24]. Another prospective follow-up study by Zhao et al. of a group of 83 patients who underwent a laparoscopic vaginoplasty with a single peritoneal graft showed a mean vaginal length of 8.2 ± 0.8 cm at 6 months and functional success at 12 months in 95.3% of the patients evaluated by the FSFI score [25]. No surgical procedure will be successful without constant vaginal dilation, either coitus or self-dilation.

The therapeutic decision tree recommends the Davydov method in cases of vaginal agenesis or patients with a previous neovaginoplasty, and the Vecchietti method when there is a vaginal dimple, in patients without a previous neovaginoplasty, and when conservative vaginal dilation procedures are ineffective [22].

Table 4 compares the main vaginoplasty techniques currently used, analyzing the type of graft, the vaginal length at 12 months, the operative time, the duration of hospitalization, the postoperative complications, and the mean FSFI score at 12 months.

The study by Callens et al. revealed that surgical success is achieved when the minimum length of the neovagina exceeds ≥7 cm, when vaginal touch easily allows for the insertion of two fingers, and when patients find sexual intercourse satisfactory [19]. Kang et al., in a cross-sectional study of 133 MRKH patients, observed that the length of the neovagina is significantly shorter in the conservative group (6.5 ± 2.04 cm) compared to the surgical group (8.1 ± 1.59 cm), and the FSFI scores were similar, except for the orgasm score, which was higher in the non-surgical group [11]. In contrast, Dabaghi et al. showed that an increased length of the created neovagina and a high degree of sexual satisfaction a observed in the surgical group compared to the conservative one [27].

In vaginoplasty procedures with a peritoneal flap, the squamous metaplasia process begins early upon contact with the external environment, creating vaginal sensitivity and elasticity corresponding to normal sexual activity [25]. On the other hand, the formation of granulation tissue and the tendency to stenosis represent major complications of techniques using peritoneal grafts. The risk of a squamous metaplastic neovaginal mucosa malignancy after vaginoplasty is low, independent of the chosen procedure. As a result, it is recommended that patients with MRKH syndrome undergo an annual gynecological examination, ±HPV testing, and/or Pap testing. However, of the 20 cases described in the literature, only 10% occurred in patients who underwent a peritoneal graft vaginoplasty procedure [28].

The main disadvantages of these surgical procedures are a short neovagina, risk of stenosis in the upper third of the neovagina, risk of prolapse over time, or, rarely, risk of evisceration. The most used procedure currently for patients with MRKH syndrome is the classic Davydov procedure, or a modified one by adapting the degree of traction of the pelvic peritoneum [29,30]. Thus, the Davydov procedure and the modified procedures of this technique have a good safety and efficiency profile compared to other surgical techniques [12]. Neovaginoplasty techniques that place additional traction on the adjacent tissues may predispose patients to an increased rate of vaginal vault prolapse. In addition to the risk of neovaginal vault prolapse, another possible risk is damage to the adjacent organs (kidneys, ureters, and rectum) in cases of an association with other congenital anomalies (ectopic kidneys and anorectal anomalies) [26]. The lowest rate of neovaginal vault prolapse have been observed after using the Davydov and Vecchietti methods [18]. Currently, no technique has been used to prevent neovaginal vault prolapse; the only cases described are surgical corrections after neovaginoplasty procedures. Studies have shown that neovaginal vault prolapse can occur after any neovaginoplasty, with a higher incidence with the intestinal graft technique [14,31].

Prolapse in patients with MRKH syndrome may result primarily from the original vaginal vault [32], or secondarily from a neovagina created by conservative or surgical procedures [14]. The possible mechanism of a prolapse is based on the lack of mechanical support at the apex and lateral walls of the created neovagina. As a result, the initial sacrocolpopexy is a surgical procedure for the prophylaxis of a prolapse of the neovagina vault, and the secondary one is for the correction of the prolapse.

Patients with MRKH syndrome are at risk of a vaginal vault prolapse following neovaginoplasty procedures. A postoperative assessment of the risk of prolapse of neovaginoplasty procedures using the POPDI-6 (Pelvic Organ Prolapse Distress Inventory) score identified the lowest value for the Williams technique, 5.6 for the McIndoe and Bowel techniques, 11.1 for the Davydov technique, and the highest score was 19.4 for the Vecchietti technique [14].

The removal of the rudimentary uterus in cases of periodic abdominal pain or obstructive bleeding is indicated in approximately 10% of patients with MRKH syndrome [6].

The presented technique consists of a perineal approach to perform a vaginoplasty followed by a laparoscopic sacropexy of the neovaginal apex, which has as its main advantage the creation of a functional neovagina of adequate size with a low risk of prolapse. This proposed vaginoplasty procedure allows patients with type I MRKH syndrome to obtain, after the intervention, a functional neovagina with an average length of 11.33 cm. The main disadvantage is the risk of erosion due to the synthetic mesh [33], which is solved either by creating a retroperitoneal tunnel or using a fragment of the autologous fascia lata graft to anchor the neovagina. An important moment in the operation is that of mobilizing and preparing the peritoneal flaps through the progressive traction of the peritoneum up to the vaginal introit.

Another potential risk is bleeding secondary to anchoring the mesh at the level of the presacral periosteum by injuring the presacral artery. Extremely rarely, processes of localized osteitis can occur [34]. The formation of granulation tissue was observed in only one patient during the first 2 months postoperatively, and was treated with vaginal estrogen. Also, the presence of postoperative granulation tissue until the epithelization of the neovaginal mucosa prevents the existence of satisfactory sexual relations. Furthermore, Origoni et al. observed that for at least 6 months postoperatively, the neovaginal epithelium showed morphological and ultrastructural similarities with the native mucosa of the vagina [35].

The early initiation of sexual relations and counseling on the use of coital dilation by these patients led to good anatomical, functional, and cosmetic results, and a low rate of complications. We also encountered these findings in the study by Herlin and his collaborators, who observed a greater length of the vagina compared to other treatment methods [36].

Postoperatively, to consolidate the results obtained in the long term (adequate length of the neovagina and a high degree of sexual satisfaction), the use of dilators or the promotion of relatively constant sexual activity is recommended. Another study showed that in the absence of sexual activity, the use of dilators, or granulation tissue, the length of the neovagina remained unchanged at approximately 9.5 cm [37].

In addition to treatment aimed at correcting vaginal agenesis, patients, most commonly in their adolescence, require careful psychological counseling for emotional problems, for the inability to have normal sexual intercourse, and for the infertility generated by this congenital anomaly [38]. Furthermore, a psychosexual education intervention more than 6 months before the intervention over 8 weeks via e-learning among a group of 38 women with MRKH syndrome observed an improvement in sexual function, genital self-image, and a decrease in sexual distress [39].

Long-term patient follow-up guarantees the success of this method, which requires the allocation of resources from public health services and the involvement of a multidisciplinary team of gynecologists, surgeons, and psychologists.

The option of a uterine transplantation in patients with MRKH syndrome after the creation of the neovagina who want children requires careful counseling and a corresponding anastomosis at the level of the neovagina. For the subsequent achievement of a uterine transplant, the creation of a sufficiently long neovagina and the absence of previous intra-abdominal neovaginoplasty surgeries are necessary, conditions fulfilled by the laparoscopic Vecchietti technique, conservative procedures, and Wharton–Sheares–George vaginoplasty [40]. In contrast, the neovagina sacropexy procedure presents superior functional results, possibly making the subsequent option of uterine transplantation more difficult, requiring careful initial counseling of these patients regarding their reproductive future [41].

The main limitation of this study is the small number of patients, with follow-up being conducted over a period of 5 years. At each medical visit, the dimensions (length and width), morphological characteristics (sensitivity), and possible complications (neovaginal stenosis, fistulas, and granulation tissue) were monitored. The good anatomical and functional results are the strength points of this study. The advantage of this surgical procedure is that postoperatively the length of the neovagina decreases only by 20–33%, without any vaginal vault prolapse risk.

Choosing a treatment plan for patients with MRKH syndrome requires counseling (Figure 3). Since currently there are many vaginal reconstruction procedures, without being able to identify the superiority of one method in relation to another, the adequate counseling of patients with MRKH syndrome regarding the choice of the best surgical option is necessary. The creation of guidelines and a unified therapeutic strategy is especially important for young patients who also want the scenario of having a pregnancy in addition to a normal sex life with the creation of a neovagina.

## 5. Conclusions

A laparoscopic peritoneal flap neovaginoplasty procedure combined with a neovaginal vault sacropexy is an effective surgical treatment option for patients with MRKH syndrome, with excellent anatomical and functional outcomes, a low risk of prolapse, and very good long-term psychosexual outcomes. Based on the three cases, this preliminary report provides evidence of a durable outcome and good quality of life compared to other surgical options for creating a neovagina.

## Figures and Tables

**Figure 1 jcm-14-01136-f001:**
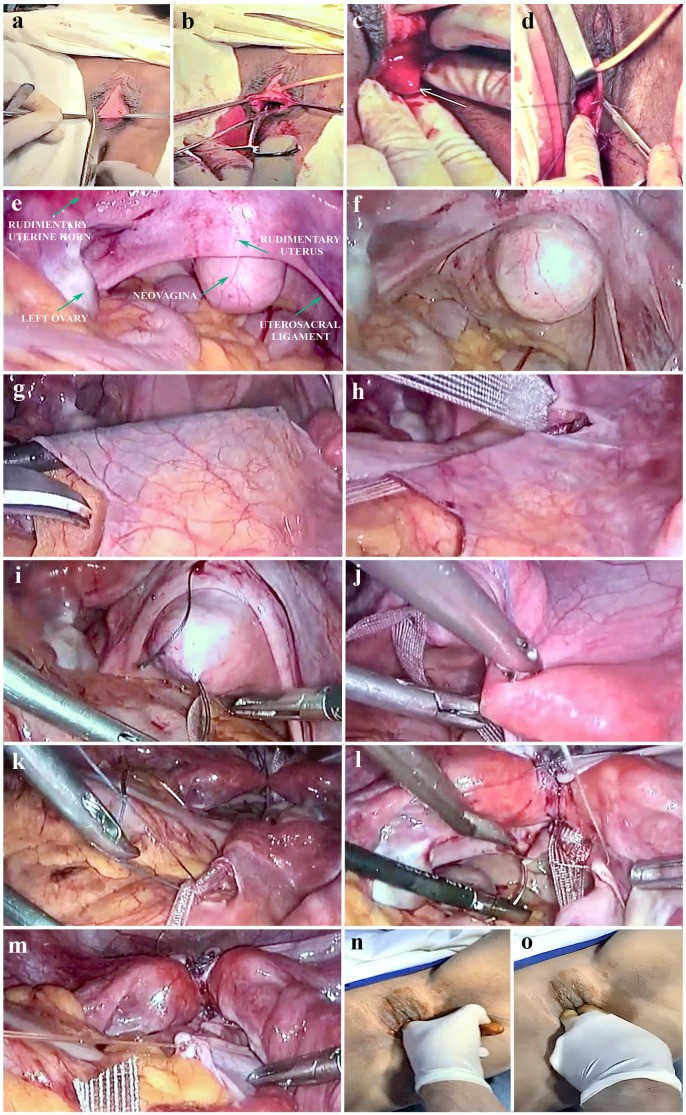
(**a**) Transverse incision of the hymen; (**b**) blunt dissection between the bladder and the rectum; (**c**) peritoneum is pulled out from the pouch of Douglas; (**d**) anchoring of the peritoneum; (**e**) laparoscopic approach—anatomical inventory of a patient with MRKH syndrome; (**f**) neovagina; (**g**) creating the retroperitoneal tunnel; (**h**) mesh introduction and placement in the tunnel; (**i**) fixation of the mesh on the rudimentary uterus and neovagina; (**j**) anchoring the round ligaments to the mesh; (**k**) fixation of the mesh on the promontory; (**l**) anchoring the uterosacral ligaments to the mesh; (**m**) closure of the median peritoneum; and (**n**,**o**) cosmetic and functional neovagina.

**Figure 2 jcm-14-01136-f002:**
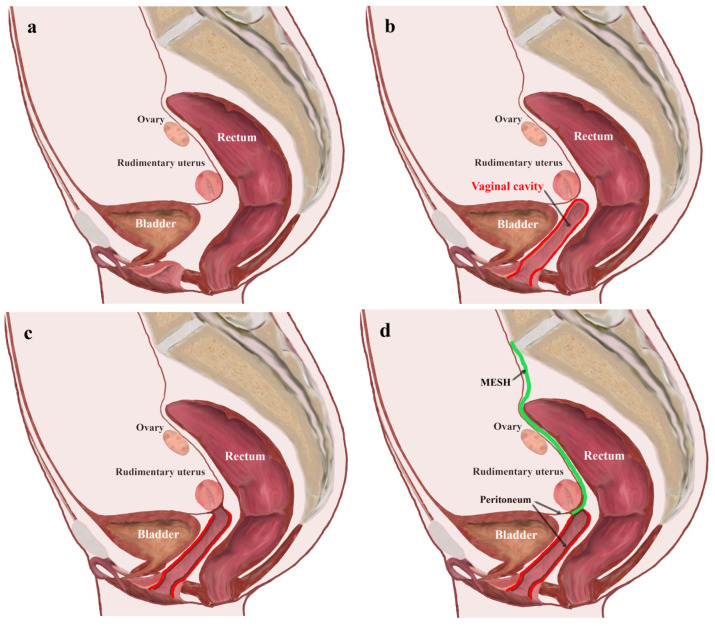
Outline of the steps of the surgical technique. (**a**) Anatomical topography of reproductive organs of patients with MRKH syndrome; (**b**) creation of the vaginal cavity in the rectovesical space; (**c**) creation of the neovagina with peritoneal flaps; (**d**) sacropexy of the neovaginal apex with mesh.

**Figure 3 jcm-14-01136-f003:**
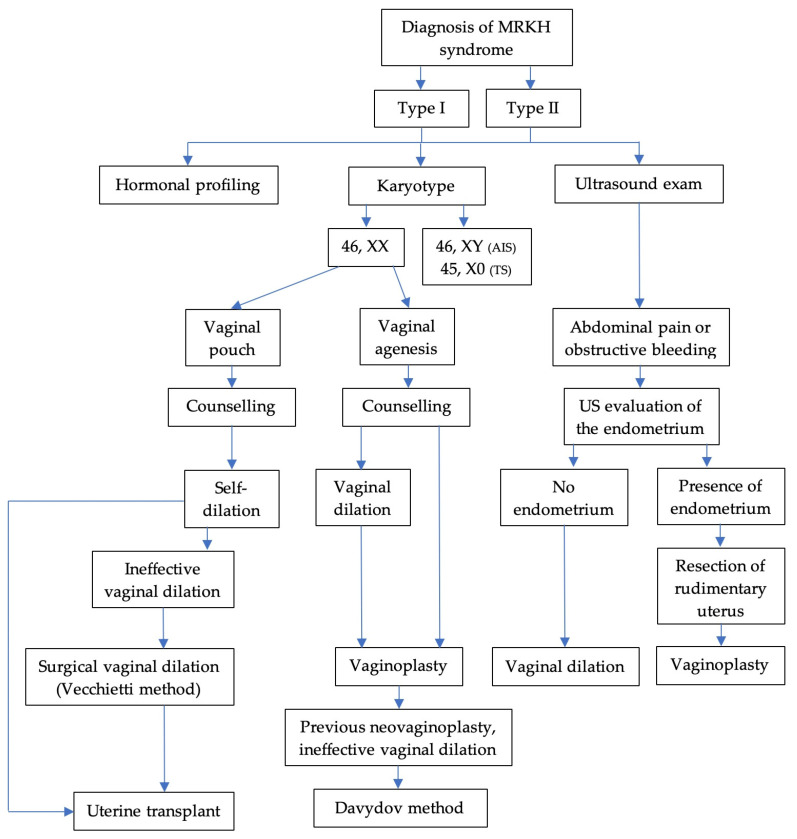
The treatment plan for patients with MRKH syndrome (AIS—androgen insensitivity syndrome; TS—Turner syndrome).

**Table 1 jcm-14-01136-t001:** Ten-step surgical algorithm for perineal vaginoplasty with a laparoscopic approach.

**Step 1**	**Perineal approach**—transverse incision of the hymen (Figure 1a). After vesical catheterization, make a 3 cm transversal hymeneal incision. With the help of Cheron forceps, a 10 cm long vaginal cavity is created by sharp, blunt dissection between the bladder and the rectum, using a compress moistened with saline solution (Figure 1b).
**Step 2**	**Anchoring of the peritoneum for the establishment of anatomical landmarks.** The peritoneum is pulled out from the pouch of Douglas through the previously created vaginal cavity (Figure 1c). The peritoneum and vaginal epithelium are anchored using absorbable 2-0 Vicryl sutures to mark and establish the anatomical landmarks (Figure 1d).
**Step 3**	**Laparoscopic approach**—placing the patient in the lithotomy position. The procedure starts with systematic control of the peritoneal cavity. The surgeon identifies the rudimentary uterine horns and the ovaries (Figure 1e,f).
**Step 4**	**Creating the retroperitoneal tunnel from the promontory to the pouch of Douglas** using monopolar scissors (Figure 1g).
**Step 5**	**Polypropylene mesh introduction into the peritoneal cavity through the trocars and placed under the previously created tunnel** (Figure 1h). The device used for neovaginal sacropexy was a polypropylene monofilament mesh (GYNECARE GYNEMESH® PS Nonabsorbable PROLENE® Soft Mesh, Ethicon, US), commonly used for pelvic floor repairs.
**Step 6**	**Fixation of the mesh on the rudimentary uterus and neovagina** using 2-0 Ethibon sutures (Figure 1i).
**Step 7**	**Anchoring the round ligaments to the mesh** (Figure 1j).
**Step 8**	**Fixation of the mesh on the promontory** using non-absorbable sutures (Figure 1k).
**Step 9**	**Anchoring the uterosacral ligaments to the mesh** (Figure 1l).
**Step 10**	**Closure of the median peritoneum** using 2-0 Ethibon sutures (Figure 1m). At the end of the procedure, use condoms filled with compresses for vaginal dilatation. The condoms need to be changed every day for 15 days. The patient is recommended to start sexual intercourse at the end of the 15 days (Figure 1n,o).

**Table 2 jcm-14-01136-t002:** Summary of clinical findings of patients.

Case	Age	AMH (ng/mL)	Genetic Analysis	Vaginal Length at Diagnosis *	Associated Malformations	Intra- and Postoperative Complications	Neovaginal Length After Surgery *	Neovaginal Length After 12 Months *	Sexual Activity (FSFI Total Score)	Quality of Life	Marital Status
1	22	6.2	46, XX	Absent	None	None	15	12	Satisfactory (26)	Good	Yes
2	24	4.3	46, XX	Absent	None	None	14	12	Satisfactory (27)	Good	No
3	25	5.2	46, XX	1	None	None	15	10	Satisfactory (26)	Good	Yes

*—centimeter (cm).

**Table 3 jcm-14-01136-t003:** Postoperative evaluation of Female Sexual Function Index (FSFI) scores among 3 cases.

	Case 1	Case 2	Case 3
**Arousal**	5	4	5
**Desire**	5	5	4
**Lubrication**	4	5	5
**Orgasm**	4	4	3
**Pain**	3	4	5
**Satisfaction**	5	5	4
**FSFI score**	26	27	26

**Table 4 jcm-14-01136-t004:** Main characteristics of different surgical vaginoplasty procedures in patients with MRKH syndrome in relation to the randomized control trial (RCT) and systematic review/meta-analysis (SR/MA) [26].

Surgical Procedures	Study Design	Type of Grafts	Vaginal Length at 12 Months (cm)	Mean FSFI Score at 12 Months	Estimated Blood Loss (mL)	Operative Time (min)	Postoperative Hospital Stay (Days)	Postoperative Complications
Davydov procedure [26]	SR/MA	peritoneum	8.3	28.9	NA	126	1–18	Stenosis (3–15%), bladder/rectum injury, ascending infection
Vecchietti vaginoplasty [26]	SR/MA	none	8.7	27.5	NA	40	2–8	Bladder/rectum injury
Peritoneal vaginoplasty [24]	RCT	peritoneum	11.89 ± 1.16	25.29 ± 0.51	65 (50–100)	100.4 ± 36.2	6 (5–7.25)	NA
Sigmoid vaginoplasty [24]	RCT	sigmoid	12.56 ± 0.42	24.79 ± 1.17	200 (100–300)	217.8 ± 40.7	10.5 (6.75–12.5)	Shrinkage of the graft, prolapse of the neovagina
Single peritoneal flap (SPF) vaginoplasty [25]	RCT	peritoneum	8.1 ± 1.3	95.3%	88.5 ± 57.2	71.2 ± 18.9	3.9 ± 1.8 (3–8)	Fibrotic stenosis (8.4%)

NA—not available.

## Data Availability

The original contributions presented in the study are included in the article, further inquiries can be directed to the corresponding authors.

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
