# Peer review of "A New Ten-Step Surgical Approach to Mayer–Rokitansky–Küster–Hauser Syndrome—A Preliminary Report of Three Cases"

_jcm, 2025, doi:10.3390/jcm14041136_

Round 1
Reviewer 1 Report
Comments and Suggestions for Authors
Dear Authors,
The presented study tackles the issue of surgical approach in Mayer-Rokitansky-Küster- Hauser syndrome. The authors present a very interesting surgical method of treatment. The study was conducted reliably with an appropriate selection of methods. Overall, I think that this article should be resubmitted after major editorial corrections stated below:
1. I suggest deleting the number with the headings in the abstract section
2. Verse 34 and 35- please add references for these descriptions
3. Verse 37- please add a reference for this statement
4. Verse 40-42- please add references for this statement
5. I suggest inserting paragraph 2 (counselling) later in the Introduction and starting with the definition of MRKH syndrome, types of MRKH, and prevalence of the disease, then going to the quality of life and methods of treatment
6. Verse 46-49 – Are you sure about this definition? Please double-check it and expand it.
7. Verse 60-64? If it’s your authorship procedure, please specify what is different in this procedure from the others.
8. Can you explain why the surgery has been performed in Morocco and all the authors are from Romania?
9. Verse 74-75 What do you mean by “No sex steroid hormone levels or chromosomal abnormality was observed in these patients.”- MRKH patients should have normal sex steroid hormone levels- please make it clear
10. Verses 76-78 are more suitable for the discussion section
11. Verse 80-81 Do you mean that the follow-up visits were scheduled every 3 months in the first year after surgery and then every 6 months in the second year? And what about the next follow-ups? You reported that the surgeries were 7-8 years ago. There is no information on the longitudinal effect and timing of follow-up visits.
12. Verse 90- shouldn’t it be presented in the past tense?
13. Can you explain why you haven’t removed the rudimentary uterus as it’s usually done because of the risk of cancer and pain during menstruation?
14. Step 1- what do you mean by annexes? Do you mean ovaries?
15. Step-2 Is the transverse incision from a vaginal or perineal approach?
16. Verse 137-139 “The advantage of this surgical procedure is the fact that the length of the neovagina remains almost the same postoperatively, without recording the prolapse of the vaginal vault.” This is more suitable for the discussion section.
17. Verse 144-145 “The advantage of this surgical procedure is the fact that the length of the neovagina remains almost the same postoperatively, without recording the prolapse of the vaginal vault.” This is more suitable for discussion.
18. Figure 2- It’s better to present the pictures in the correct order e.g. firstly you created neovagina to visualize it in the laparoscopy, therefore I suggest putting pictures a and b after f.
19. What about the results of FSFI? It’s worth presenting it.
20. Verse 168-171 is more suitable for the introduction.
21. Verse 172-179 and what are the results of these researches?
22. Verses 185-188 I suggest rephrasing this sentence.
23. Verse 189 This part should be with the other sentences tackling counseling issues
24. The discussion section is very chaotic. Please try to systemize the information eg starting from general information on the possible options of treatment (conservative and surgical methods) and then going to particular pros and cons of each, then finishing on the proposed by you with pros and cons.
25. Verse 213-222 please add reference for these complications.
Comments on the Quality of English LanguageThe English Language should be improved.
Author Response
Dear Reviewer,
Thank you for taking the time to read our article and suggesting valuable changes to improve the quality of the manuscript. Please find the detailed responses below and the corresponding revisions by section/line number in the resubmitted files.
- I suggest deleting the number with the headings in the abstract section
Response: The authors agree that the numbers can be deleted. Please see the attached manuscript.
- Verse 34 and 35- please add references for these descriptions
Response: Thank you for suggesting this. We added references. Please see lines 35-36.
- Verse 37- please add a reference for this statement
Response: Thank you for your suggestion. We added references. Please see lines 101-102.
- Verse 40-42- please add references for this statement
Response: Thank you for your remark. We added references. Please see lines 94-96.
- I suggest inserting paragraph 2 (counselling) later in the Introduction and starting with the definition of MRKH syndrome, types of MRKH, and prevalence of the disease, then going to the quality of life and methods of treatment
Response: Thank you for your valuable remark. We agree with this, so we have started the introduction as you suggested and moved the paragraph about counseling later in the text. Please see the attached manuscript lines 33-41, 94-96)
- Verse 46-49 – Are you sure about this definition? Please double-check it and expand it.
Response: Thank you for your remark. We checked the definition and corrected it. Please see lines 42-46.
- Verse 60-64? If it’s your authorship procedure, please specify what is different in this procedure from the others.
Response: Unlike other surgical techniques, our surgical procedure addresses the prophylaxis of urinary and pelvic floor symptoms (neovaginal apex prolapse). The presented laparoscopic procedure derives from the Davydov technique. Please see lines 87-92.
- Can you explain why the surgery has been performed in Morocco and all the authors are from Romania?
Response: Two of the authors are from Romania, and three are from Morocco. Doctor Yassin completed his residency training in Romania. After that, we continued to collaborate through the exchange of experience between clinics, including video participation at different surgeries to learn new techniques.
- Verse 74-75 What do you mean by “No sex steroid hormone levels or chromosomal abnormality was observed in these patients.”- MRKH patients should have normal sex steroid hormone levels- please make it clear
Response: Thank you for pointing this out. It was a typo. We meant that no imbalance in sex steroid hormone levels was observed. Please see lines 113-114.
- Verses 76-78 are more suitable for the discussion section
Response: Thank you for your suggestion. The authors agree that this information should be moved to the discussion section. Please see lines 382-387.
- Verse 80-81 Do you mean that the follow-up visits were scheduled every 3 months in the first year after surgery and then every 6 months in the second year? And what about the next follow-ups? You reported that the surgeries were 7-8 years ago. There is no information on the longitudinal effect and timing of follow-up visits.
Response: These patients were followed up after surgery every 3 months for the first year and then every 6 months for the next 5 years, after which they did not return to the clinic. Please see lines 202-205.
- Verse 90- shouldn’t it be presented in the past tense?
Response: Thank you for your remark. We modified the tense. Please see lines 167-169.
- Can you explain why you haven’t removed the rudimentary uterus as it’s usually done because of the risk of cancer and pain during menstruation?
Response: We agree that the rudimentary uterus could have been excised. In the cases presented, the patients did not menstruate prior to surgery, and ultrasound evaluation did not reveal an endometrial cavity. Thus, as the patients were asymptomatic, it was decided to preserve the rudimentary uterus and use it for mesh fixation.
- Step 1- what do you mean by annexes? Do you mean ovaries?
Response: Thank you for your observation. It was an error because we meant to refer to the ovaries. Please see Table 1, Step 3.
- Step-2 Is the transverse incision from a vaginal or perineal approach?
Response: Thank you for this important remark. It is about a perineal approach. Please see Table 1 Step 1.
- Verse 137-139 “The advantage of this surgical procedure is the fact that the length of the neovagina remains almost the same postoperatively, without recording the prolapse of the vaginal vault.” This is more suitable for the discussion section.
Response: Thank you for your suggestion. We moved this information to the discussion section. Please see lines 378-380.
- Figure 2- It’s better to present the pictures in the correct order e.g. firstly you created neovagina to visualize it in the laparoscopy, therefore I suggest putting pictures a and b after f.
Response: We are grateful for this comment. As you suggested, we have rearranged the photos in Figure 1 (e.g., Figure 2). Please see the attached manuscript.
- What about the results of FSFI? It’s worth presenting it.
Response: We agree that the FSFI result should be presented. Additional discussions have been included in the manuscript. Please see Table 2 and Table 3, where FSFI scores are presented, lines 196-201.
- Verse 168-171 is more suitable for the introduction.
Response: Thank you for your suggestion. We moved the paragraph regarding the etiology of MRKH to the Introduction. Please see the lines 36-41.
- Verse 172-179 and what are the results of these researches?
Response: Thank you for your mention. We rewrote the paragraph for better understanding. Please see lines 257-268.
- Verses 185-188 I suggest rephrasing this sentence.
The parameters that allow the evaluation of surgical success, according to the study by Callens et al., are represented by a minimum postoperative length of the neovagina that exceeds ≥7 cm, vaginal dilation that easily allows the insertion of two fingers, and the patients' assessment of sexual intercourse as satisfactory.
Response: Thank you for your valuable remark. We rewrote as you suggested. Please see lines 280-282.
- Verse 189 This part should be with the other sentences tackling counseling issues.
Response: Thank you for your suggestion. We moved it along with the other data related to counseling. Please see lines 381-387 and Figure 3.
- The discussion section is very chaotic. Please try to systemize the information eg starting from general information on the possible options of treatment (conservative and surgical methods) and then going to particular pros and cons of each, then finishing on the proposed by you with pros and cons.
Response: The discussion section has been rewritten and rearranged for better understanding. Thank you for pointing out these discrepancies. Please see the attached manuscript.
- Verse 213-222, please add a reference for these complications.
Response: Thank you for this mention. We included references. Please see lines 330-343
Reviewer 2 Report
Comments and Suggestions for Authors
This is an interesting manuscript presenting original operative approach in MRKH patients. There are some serious flaws of this study...
In the title it should be mentioned that this is presentation of 3 cases... So initial or preliminary report; an attempt of introduction of a new method... (?)
In the introduction there is missing presentation of alternatives...
Figure 1 is not cited sufficiently and properly in the text.
Table 2 have numbers which do not correlate with the figure 2. Should be revised.
Table 3 does not present comprehensive data as this are just separate studies without metaanalysis... It cannot be correlated or even analyzed with results of this report as this are data of single, non-randomized studies... Only results of metaanalysis may be placed in such table and it is strongly suggested to do so...
Reference 5 and 12 are the same.
Conclusions are rather repeated results than real conclusions, to be reformulated.
There are two possible way of revision this report:
1. To explain weak and strong sides of introduced method in regard to explanation in the discussion part step by step already used methods.
2. To perform metaanalysis of surgical treatment methods and to compare them in table with introduced method (10-step approach technique).
Author Response
Dear Reviewer,
Thank you for taking the time to read our article and suggesting valuable changes to improve the quality of the manuscript. Please find the detailed responses below and the corresponding revisions by section/line number in the resubmitted files.
In the title it should be mentioned that this is presentation of 3 cases... So initial or preliminary report; an attempt of introduction of a new method... (?)
Response: Thank you for pointing out this. We modified the title as you suggested. Please see the attached manuscript.
In the introduction there is missing presentation of alternatives...
Response: Thank you for your valuable remark. We inserted the alternative procedure to surgery in the manuscript. Please see lines 67-69.
Figure 1 is not cited sufficiently and properly in the text.
Response: Thank you for your valuable remark. We modified the citations. Please see Figure 2 (ex-figure 1) in the attached manuscript.
Table 2 have numbers which do not correlate with the figure 2. Should be revised.
Response: Thank you for pointing out this discrepancy. We have corrected the error. Please see Table 1 in the attached manuscript.
Table 3 does not present comprehensive data as this are just separate studies without metaanalysis... It cannot be correlated or even analyzed with results of this report as this are data of single, non-randomized studies... Only results of metaanalysis may be placed in such table and it is strongly suggested to do so...
Response: Thank you for this valuable remark. We have reworked the table. You can find it in the revised manuscript as Table 4, in which we presented the main characteristics of the different surgical vaginoplasty procedures in patients with MRKH syndrome in relation to the randomized control trial (RCT) and the systematic review/meta-analysis (SR/MA).
References 5 and 12 are the same.
Response: Thank you for alerting us. We deleted the duplicate reference. Please see the attached manuscript.
Conclusions are rather repeated results than real conclusions, to be reformulated.
Response: Thank you for your suggestion. We have reformulated the conclusions.
There are two possible way of revision this report:
- To explain weak and strong sides of introduced method in regard to explanation in the discussion part step by step already used methods.
- To perform metaanalysis of surgical treatment methods and to compare them in table with introduced method (10-step approach technique).
Response: Thank you for your suggestions. We chose to explain the advantages and disadvantages of our technique compared to those available in the discussion section and created Table 4 for better systematization of the data.
Round 2
Reviewer 1 Report
Comments and Suggestions for Authors
The authors addressed all my suggestions. The only suggestion is to insert the last figure in the manuscript, not after the conclusions.
Reviewer 2 Report
Comments and Suggestions for Authors
I see that the Authors introduced amendments and sufficiently improved their paper according to given remarks. I don't have any further suggestions and feel convinced to recommend revised paper for publication.